# Outcomes of Rural Men with Breast Cancer: A Multicenter Population Based Retrospective Cohort Study

Lucas A. B. Fisher [1], Osama Ahmed [1,2], Haji Ibraheem Chalchal [1,3], Ray Deobald [4], Ali El-Gayed [1,5], Peter Graham [4], Gary Groot [4], Kamal Haider [1,2], Nayyer Iqbal [1,2], Kate Johnson [1,5], Duc Le [1,5], Shazia Mahmood [1,6], Mita Manna [1,2], Pamela Meiers [4], Mehrnoosh Pauls [7], Muhammad Salim [1,3], Amer Sami [1,2], Philip Wright [1,5], Moftah Younis [1,5] and Shahid Ahmed [1,2,*]

1   Division of Oncology, College of Medicine, University of Saskatchewan, Saskatoon, SK S7N 5E5, Canada
2   Medical Oncology, Saskatoon Cancer Center, Saskatchewan Cancer Agency, Saskatoon, SK S7N 4H4, Canada
3   Medical Oncology, Allan Blair Cancer Center, Saskatchewan Cancer Agency, Regina, SK S4T 7T1, Canada
4   Department of Surgery, University of Saskatchewan, Saskatoon, SK S7N 0W8, Canada
5   Radiation Oncology, Saskatoon Cancer Center, Saskatchewan Cancer Agency, Saskatoon, SK S7N 4H4, Canada
6   Radiation Oncology, Allan Blair Cancer Center, Saskatchewan Cancer Agency, Regina, SK S4T 7T1, Canada
7   BC Cancer Agency, Vancouver, BC V5Z 1G1, Canada
*  Correspondence: shahid.ahmed@saskcancer.ca; Tel.: +30-66-552-710; Fax: +30-66-550-633

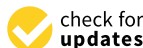

**Simple Summary:** Breast cancer is rare in men. This study compared the outcomes of rural men to urban men who were diagnosed with breast cancer over a 20-year period in the province of Saskatchewan, Canada. Most men were diagnosed with hormone receptor-positive early-stage breast cancer. Although rural men numerically had shorter survival, it was not statistically significant. Stage of the disease, poor performance status, node-positive disease, older age, and lack of adjuvant systemic therapy were associated with poor outcomes. Adjuvant therapy should be offered to men with breast cancer to reduce the risk of recurrent breast cancer. Future research is warranted to determine whether the numerical differences seen in rural residence survival are observed in a larger patient population.

**Abstract:** Background: Breast cancer is rare in men. This population-based study aimed to determine outcomes of male breast cancer in relation to residence and other variables. Methods: In this retrospective cohort study, men diagnosed with breast cancer in Saskatchewan during 2000–2019 were evaluated. Cox proportional multivariable regression analyses were performed to determine the correlation between survival and clinicopathological and contextual factors. Results: One hundred-eight eligible patients with a median age of 69 years were identified. Of them, 16% had WHO performance status $\geq$ 2 and 61% were rural residents. The stage at diagnosis was as follows: stage 0, 7%; I, 31%; II, 42%; III, 11%; IV, 8%. Ninety-eight percent had hormone receptor-positive breast cancer. The median disease-free survival of urban patients was 97 (95% CI: 50–143) vs. 64 (46–82) months of rural patients ($p$ = 0.29). The median OS of urban patients was 127 (94–159) vs. 93 (32–153) months for rural patients ($p$ = 0.27). On multivariable analysis, performance status $\geq$ 2, hazard ratio (HR) 2.82 (1.14–6.94), lack of adjuvant systemic therapy, HR 2.47 (1.03–5.92), and node-positive disease, HR 2.32 (1.22–4.40) were significantly correlated with inferior disease-free survival in early-stage invasive breast cancer. Whereas stage IV disease, HR 7.8 (3.1–19.5), performance status $\geq$ 2, HR 3.25 (1.57–6.71), and age $\geq$ 65 years, HR 2.37 (1.13–5.0) were correlated with inferior overall survival in all stages. Conclusions: Although residence was not significantly correlated with outcomes, rural men had numerically inferior survival. Poor performance status, node-positive disease, and lack of adjuvant systemic therapy were correlated with inferior disease-free survival.

**Keywords:** male breast cancer; rural residence; survival; cohort; breast malignancy; breast cancer; men; systemic therapy; performance status; age

## 1. Introduction

Breast cancer is a heterogeneous disease that is broadly categorized into hormone receptor-positive breast cancer, HER2-positive breast cancer, and triple-negative breast cancer. It is the most common cancer diagnosed in women, with more than 27,000 being diagnosed each year in Canada [1]. In 2020, the worldwide incidence of breast cancer was approximately 2.3 million [2]. It remains a leading cause of cancer-related death in women [3,4]. However, breast cancer in men is much rarer and accounts only for 1% of all breast cancer cases and <1% of all cancers in men. Due to the sparsity of cases, there is a paucity of prospective studies and randomized clinical trials investigating male breast cancer. Most information on the management of male breast cancer has been extrapolated from randomized trials conducted in women.

The majority of breast cancer cases in men are symptomatic at the time of diagnosis. It is important to know that male breast cancer is usually diagnosed at a later stage and is often accompanied by more advanced disease features [5]. Node-positive breast cancer at the time of presentation is more common in men than women [6]. Additionally, male breast cancer cases are almost entirely hormone receptor-positive with roughly 90% of cases showing receptor positivity, which is not the case for female breast cancer, showing a hormone receptor positivity rate of about 70% [7]. Male breast cancer also has been shown to differ from female breast cancer in rates of histological subtypes and molecular changes, including genomic, transcriptional, and expression profiles [8]. Men more often have an underlying germline cancer predisposition and are at greater risk of developing contralateral breast cancer than women [7]. Thus, the uniqueness of male breast cancer warrants further studies examining treatment and outcomes specific to this patient population.

Over the past two decades, there have been large improvements in the diagnosis and treatment of breast cancer [9,10]. Surgical treatment and radiation therapy for early-stage breast cancer have evolved, reducing long-term cosmetic and functional sequelae [10]. Similarly, access to more effective adjuvant therapy in early-stage disease, and to novel systemic therapies such as CDK4/6 for advanced disease, have improved outcomes for both early and advanced-stage breast cancer [9].

Despite advancements in the management of breast cancer, disparities in cancer incidence and outcomes have been observed between urban and rural patients with several malignancies including breast cancer [11,12]. Several investigators have reported that rural breast cancer populations have differential incidence and outcomes compared to their urban counterparts [11–15]. Lower use of screening mammography, difficulty accessing primary physician or specialist care, as well as impaired access to treatments due to long travel times in rural breast cancer patients, may affect their outcomes [16,17]. To our knowledge, no study has specifically examined the outcomes of rural male breast cancer patients. The current study aimed to examine outcomes of male breast cancer over the past two decades in Saskatchewan. Saskatchewan, a central Canadian province, has a large rural population that provides an opportunity to examine the outcomes of rural male breast cancer patients.

## 2. Methods

### 2.1. Study Population

The University of Saskatchewan Biomedical Research Ethics Board (Bio-REB) approved the study (Application ID: 3139). In addition, operational approval was obtained from the Data Access Committee of the Saskatchewan Cancer Agency (Application ID: 1-21-016). This is a retrospective cohort study that involves men with breast carcinoma diagnosed in the province of Saskatchewan, Canada, from January 2000 to December 2019. Patients with neuroendocrine breast cancer, lymphoma, sarcoma, melanoma, and desmoid tumors were excluded. In addition, patients with another active invasive cancer, those who moved to other provinces, or those who died within 4 weeks of diagnosis of breast cancer were excluded. The Saskatchewan Cancer Registry which prospectively collects information on all cancers was used to identify eligible patients. International Classification of Disease (ICD) codes relevant to breast cancer were used to identify eligible patients. In

addition to registry data, individual medical records were reviewed by the study team members, and study-related information was collected.

### 2.2. Definitions

The Statistics Canada definition of rural towns—less than 50,000 population [18]—was used for defining the rural location of the patient. Overall survival (OS) was defined as the time from the diagnosis of breast cancer to death from any cause. Patients who were alive at the end of the follow-up period or were lost to follow-up during the study were censored at the last date they were known to be alive. Disease-free survival (DFS) was defined as the time from the date of surgery of early-stage operable breast cancer to the date of relapse, secondary cancer, or death from any cause. Relapse-free survival (RFS) was defined as the time from the date of surgery of early-stage operable breast cancer to the date of relapse or death from any cause. Major co-morbid illnesses were defined as the presence of coronary artery disease, diabetes mellitus, chronic renal insufficiency, chronic obstructive lung disease and others (uncontrolled hypertension, peripheral vascular disease, stroke or transient ischemic attack, interstitial lung disease, congestive heart failure, cardiac arrhythmia, multiple sclerosis, inflammatory bowel disease, etc.). Secondary cancer was defined as a past history of invasive cancer requiring treatment. Adjuvant systemic therapy was defined as the use of endocrine, chemotherapy, or trastuzumab alone or in different combinations with curative surgery.

### 2.3. Statistical Analysis

The chi-square and Fisher Exact tests were performed for categorical variables for $p$ values estimation. Student's $t$-test was performed for analysis of continuous variables. Survival of the entire cohort and subgroups were estimated using the Kaplan−Meier method, and the survival distribution of different groups was compared by the log rank test. The overall significance level is set at two-tailed $p$-value of 0.05.

The multivariable Cox proportional hazard models were used, and the hazard ratio (HR) and its 95% confidence limit (CI) were estimated for OS of the entire cohort and DFS of patients with early-stage invasive breast cancer (stages I, II, and III). The following variables were examined for OS and DFS: rural residence, time period (2000–2009 vs. 2010–2019), age $\geq$ 65 years, the World Health Organization (WHO) performance status $\geq$ 2, presence or absence of a major comorbid illness as per definition, HER2 positive disease, past history of a secondary cancer, neutrophil:lymphocyte > 2.5, alkaline phosphatase > 110 U/L, albumin < 35 g/L, creatinine $\geq$ 105 umol/L, hemoglobin < 135 g/L, WBC > 11 $\times$ $10^9$/L, and platelets > 400 $\times$ $10^9$/L. For early-stage invasive breast cancer, we also examined the prognostic importance of >T2 disease, node-positive disease, grade III tumor, positive surgical margin, adjuvant systemic therapy, and adjuvant radiation on DFS. For OS, we additionally examined the prognostic significance of stage IV disease. The variables with a $p$-value of <0.10 on univariate analysis or biologically important were fitted into the multivariable model. The proportional hazards assumption was assessed using log-log survival curves in the final mathematical model. All patients were followed until August 2022 when data entry was closed. SPSS version 27.0 was used for statistical analysis (IBM, Armonk, NY, USA).

## 3. Results

### 3.1. Patient Characteristics

From the registry, 118 male patients with mammary carcinoma were identified. Of them, 10 patients were excluded for the following reasons: a secondary active metastatic cancer (n = 1), other malignancies of breast (n = 3), moved to other provinces (n = 2), and died within 4 weeks of diagnosis (n = 4) (Figure 1). Baseline characteristics of the patients are shown in Table 1. The median age of the entire cohort was 69 years (30–91). Of them, 75% had a comorbid illness, 26% had a past history of a secondary cancer, and 16% had WHO performance status $\geq$ 2. The breast cancer stage at diagnosis was as follows:

stage 0, 7%; stage I disease, 31%; stage II disease, 42%; stage III disease, 11% and stage IV disease, 8%. Of all patients, 52% had node positive disease. Among all patients, 98% had hormone receptor (HR) positive breast cancer, 11% had HER2+ve disease, and only 1% had triple negative cancer. Overall, 99% of patients with early-stage disease had a mastectomy, 81% received adjuvant systemic therapy, and 32% received radiation treatment. Among 108 patients, 61% were rural residents and 42% were diagnosed prior to 2010. No significant difference was noted in the baseline characteristics and treatment between urban and rural patients. Of note, using the Saskatchewan Registry data, we identified 10,641 women who were diagnosed with invasive breast cancer between 2005 and 2017. Of them, 241 women were excluded due to unknown staging information. Among them 12.2% had stage 0 disease, 39.3% had stage I disease, 30.94% had stage II disease, 11.45% had stage III disease, and 6.22% had de novo stage IV disease.

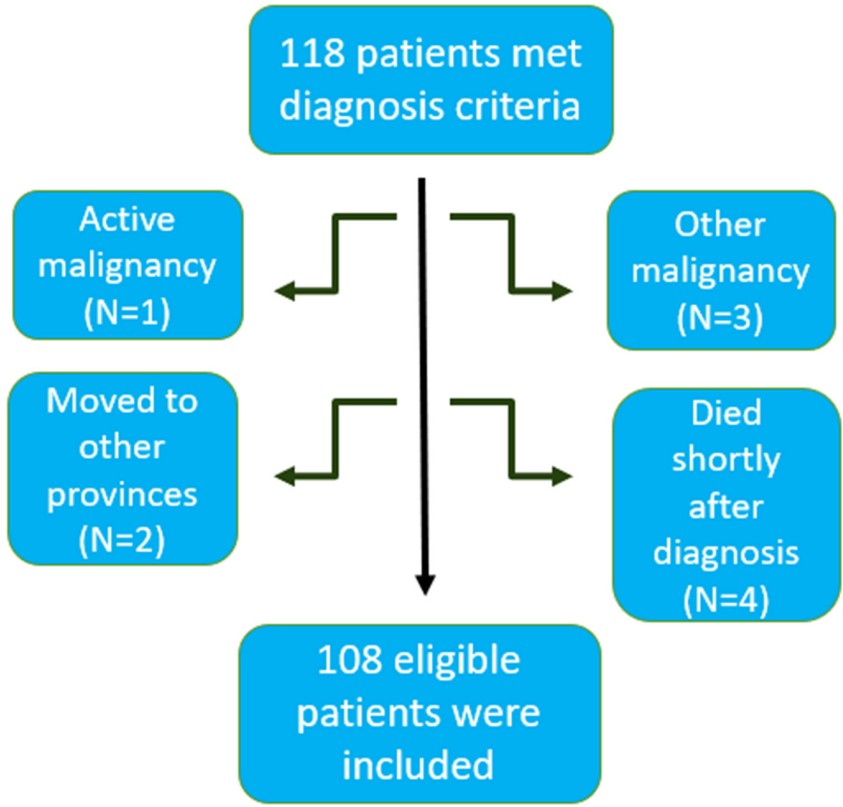

**Figure 1.** Flowchart of eligible participants evaluated.

**Table 1.** Baseline Characteristics of patients.

| Variables | All<br>N = 108 (%) | Rural<br>N = 66 (%) | Urban<br>N = 42 (%) | *p* Value |
|---|---|---|---|---|
| Median Age (range) | 69 (30–91) | 69 (30–89) | 69 (36–91) | 0.62 |
| Age ≥ 65 years | 69 (64) | 41 (62) | 28 (67) | 0.68 |
| Time period (2000–2009) | 45 (42) | 30 (45) | 15 (36) | 0.42 |
| Presence of comorbid illnesses [a] | 81 (75) | 47 (71) | 34 (81) | 0.36 |
| Past history of a secondary cancer | 27 (26) | 17 (26) | 10 (24) | 1.0 |
| WHO Performance Status [b] | | | | |

**Table 1.** *Cont.*

| Variables | All<br>N = 108 (%) | Rural<br>N = 66 (%) | Urban<br>N = 42 (%) | *p* Value |
|---|---|---|---|---|
| 0 | 58 (56) | 37 (59) | 21 (51) | 0.55 |
| 1 | 30 (29) | 17 (27) | 13 (32) | 0.66 |
| 2 | 13 (13) | 7 (11) | 6 (14) | 0.56 |
| 3 | 3 (3) | 2 (3) | 1 (2) | 1.0 |
| Active smoker | 59 (55) | 37 (56) | 22 (52) | 0.21 |
| Stage [c] | | | | |
| 0 | 8 (7) | 6 (9) | 2 (5) | 0.77 |
| I | 33 (31) | 17 (26) | 16 (38) | 0.20 |
| II | 46 (42) | 28 (42) | 18 (43) | 1.0 |
| III | 12 (11) | 10 (15) | 2 (5) | 0.12 |
| IV | 9 (8) | 5 (8) | 4 (10) | 0.73 |
| Site | | | | |
| Right breast | 48 (44) | 32 (49) | 16 (38) | 0.32 |
| Left breast | 59 (55) | 33 (50) | 26 (62) | 0.24 |
| Bilateral | 1 (1) | 1 (2) | 0 | 1.0 |
| Surgery | | | | |
| Breast conserving surgery | 1 (1) | 1 (2) | 0 | 1.0 |
| Mastectomy | 95 (88) | 58 (88) | 37 (88) | 1.0 |
| Bilateral mastectomy | 4 (4) | 4 (6) | 0 | 0.15 |
| No surgery | 8 (7) | 3 (4) | 5 (12) | 0.25 |
| Axillary nodal dissection | 66 (61) | 44 (66) | 22 (52) | 0.16 |
| Multifocal disease | 9 (8) | 4 (6) | 5 (12) | 0.30 |
| T status [d] | | | | |
| TIS | 8 (7) | 6 (9) | 2 (5) | 0.48 |
| T1 | 48 (45) | 28 (42) | 20 (49) | 0.69 |
| T2 | 35 (33) | 23 (35) | 12 (29) | 0.53 |
| T3 | 3 (3) | 2 (3) | 1 (2) | 1.0 |
| T4 | 13 (12) | 7 (11) | 6 (15) | 0.56 |
| Grade | | | | |
| I | 15 (14) | 7 (11) | 8 (19) | 0.26 |
| II | 45 (42) | 31 (48) | 14 (33) | 0.23 |
| III | 39 (36) | 24 (37) | 15 (38) | 1.0 |
| NA | 9 (8) | 4 (6) | 5 (12) | 0.31 |
| Nodal status | | | | |
| N0 | 52 (48) | 31 (47) | 21 (50) | 0.84 |
| N1 | 33 (31) | 21 (32) | 12 (29) | 0.83 |
| N2 | 9 (8) | 6 (9) | 3 (7) | 1.0 |
| N3 | 4 (4) | 3 (5) | 1 (2) | 1.0 |
| NA | 10 (9) | 5 (8) | 5 (12) | 0.51 |
| Positive resection margin | 9 (8) | 4 (6) | 5 (12) | 0.31 |

**Table 1.** *Cont.*

| Variables | All N = 108 (%) | Rural N = 66 (%) | Urban N = 42 (%) | *p* Value |
|---|---|---|---|---|
| Lymphovascular invasive | 34 (31) | 24 (36) | 10 (24) | 0.21 |
| Underlying DCIS | 35 (32) | 22 (33) | 13 (31) | 0.84 |
| Receptor status [e] | | | | |
| ER/PR positive | 98 (98) | 59 (98) | 39 (98) | 0.74 |
| HER2 positive | 12 (12) | 5 (8) | 7 (18) | 0.21 |
| Triple negative | 1 (1) | 0 | 1 (2) | 0.38 |
| Mean creatinine μmol/L | 97 ± 33 | 97 ± 36 | 97 ± 28 | 0.92 |
| Mean Albumin g/L | 38 ± 4 | 39 ± 4 | 38 ± 3 | 0.13 |
| Mean Alkaline phosphatase U/L | 86 ± 40 | 83 ± 31 | 90 ± 50 | 0.43 |
| Mean bilirubin μm/L | 10 ± 5 | 10 ± 8 | 9 ± 5 | 0.22 |
| Mean ALT U/L | 28 ± 26 | 31 ± 31 | 24 ± 15 | 0.20 |
| Mean AST U/L | 26 ± 11 | 26 ± 10 | 27 ± 14 | 0.75 |
| Mean WBC × $10^9$/L | 7.5 ± 2.0 | 7.2 ± 2.1 | 7.8 ± 1.9 | 0.17 |
| Mean hemoglobin g/L | 144 ± 18 | 146 ± 20 | 142 ± 16 | 0.18 |
| Mean platelets × $10^9$/L | 235 ± 72 | 230 ± 68 | 242 ± 78 | 0.44 |
| Median NLR (IQR) | 2.6 (2.0–3.7) | 2.9 (2.0–3.7) | 2.4 (1.9–3.5) | 0.52 |

[a] In four patients information was not available, [b] in four patients information was not available, [c] one patient stage 2 was suspected, [d] in one patient information was missing, [e] in eight patients including six patients with DCIS ER/PR status was not reported. ± standard deviation. ALT = alanine aminotransferase; AST = aspartate aminotransferase; DCIS = ductal carcinoma in situ; IQR = interquartile range; NA = not available, WBC = white blood cell; WHO = World Health Organization.

### 3.2. Intervention and Survival

The interventions and outcomes of urban and rural patients are shown in Table 2. Most patients with early-stage cancer received systemic therapy in both groups, 79% for rural patients and 87% for urban. Types of adjuvant treatment included endocrine therapy, chemotherapy with endocrine, chemotherapy alone, and trastuzumab with chemotherapy. Roughly a third of both rural and urban patients received radiation therapy (Table 2). Tamoxifen was prescribed as adjuvant endocrine therapy in 99% of cases. With respect to chemotherapy, a combination of docetaxel and cyclophosphamide for four or six cycles was used in 40% of cases and three or four cycles of 5FU, epirubicin, and cyclophosphamide (FEC-100) followed by three or four cycles of docetaxel were used in 51% of cases.

The median follow-up period for all patients was 75 months and the total follow-up duration was 240 months. There was no statistically significant difference in terms of DFS events, including recurrent breast cancer, distant metastases, new invasive breast cancer, or deaths in the two groups. Most events in patients with early-stage BC were due to death unrelated to breast cancer or the development of a secondary cancer during the follow-up period. For advanced disease, there were no statistically significant differences in treatment interventions, which included endocrine therapy, cyclin-dependent kinase 4/6 inhibitors, or chemotherapy.

Overall survival based on stage for the whole cohort is shown in Figure 2. A statistically significant difference was noted in the median OS based on the stage of the disease (*p* < 0.001). The Median OS of all patients was 114 months (70–158): stage 0 disease, 202 months; stage I disease, 188 months (95% CI: 106–271); stage II disease, 108 months (95% CI: 74–142); stage III disease, 78 months (95% CI: 75–81); stage IV disease, 14 months (95% CI: 2–26). Additionally, patients with early-stage disease (stages I–III) who did not receive adjuvant systemic therapy had a significantly lower median DFS of 43 months

(95% CI: 17–69) compared to 90 months (95% CI: 65–115) if they received adjuvant therapy ($p$ = 0.036) (Figure 3). Likewise, men with early-stage breast cancer (stage I–III) who did not receive adjuvant systematic had a median RFS of 43 months (95% CI: 17–69) compared to 126 months (77–175) if they received adjuvant systemic therapy ($p$ = 0.005). Overall, 15% of patients developed recurrent breast cancer and among them, 66% developed distant metastases. Among men with early-stage breast cancer, 20% developed a new secondary cancer and 24% died from causes unrelated to breast cancer (Table 2).

**Table 2.** Interventions and outcomes of the entire cohort and the two groups.

| Variables | All N = 108 (%) | Rural N = 66 (%) | Urban N = 42 (%) | $p$ Value |
|---|---|---|---|---|
| Median follow up in months (IQR) | 75 (39–127) | 74 (42–135) | 80 (34–118) | 0.58 |
| Received Systemic therapy for early stage disease | 81 of 99 (81) | 48 of 61 (79) | 33 of 38 (87) | 0.42 |
| Adjuvant | 73 (90) | 45 (94) | 28 (85) | 0.26 |
| Neoadjuvant | 8 (10) | 3 (6) | 5 (15) | 0.26 |
| Type of adjuvant treatment [a] | | | | |
| Endocrine therapy alone [b] | 45 (55) | 24 (50) | 21 (64) | 0.26 |
| Chemotherapy plus endocrine [c] | 33 (41) | 22 (46) | 11 (33) | 0.35 |
| Chemotherapy alone | 2 (2) | 1 (2) | 1 (3) | 1.0 |
| Trastuzumab with chemotherapy | 11 (14) | 5 (10) | 6 (18) | 0.34 |
| Completed planned chemotherapy [d] | 26 (74) | 17 (74) | 9 (75) | 1.0 |
| Adjuvant radiation | 32 of 99 (32) | 21 of 61 (34) | 11 of 38 (29) | 0.66 |
| Disease-Free Survival events | 60 of 99 (61) | 38 of 61 (62) | 22 of 38 (58) | 0.67 |
| Recurrent breast cancer | 15 (25) | 8 (21) | 7 (32) | 0.37 |
| Distant metastases | 10 (66) | 7 (88) | 3 (43) | 0.73 |
| New invasive breast cancer | 1 (2) | 0 | 1 (5) | 1.0 |
| New secondary cancer | 20 (33) | 14 (37) | 6 (27) | 0.57 |
| Death | 24 (40) | 16 (42) | 8 (36) | 0.78 |
| Advanced disease (de novo/relapse) | 19 (18) | 12 (18) | 7 (17) | 1.0 |
| Endocrine therapy | 15 (79) | 8 (67) | 7 (100) | 0.45 |
| CDK 4/6 inhibitor | 11 (58) | 4 (34) | 7 (100) | 0.45 |
| Chemotherapy for advanced disease | 4 (21) | 2 (17) | 2 (29) | 0.60 |
| Median line of therapy for advanced disease (range) | 2 (0–6) | 2 (1–5) | 2 (0–6) | 1.0 |

[a] treatments were not mutually exclusive; [b] tamoxifen was prescribed in 76 of 77 cases (99%); [c] cases most common regimen was four or six cycles of docetaxel (40%) and cyclophosphamide or three or four cycles of FEC-100 followed by three or four cycles of docetaxel (51%); [d] 14% case it was stopped due to treatment related toxicities and 11% case as per patients request. CDK = cyclin-dependent kinase; IQR = interquartile range.

Figure 4 shows comparisons of rural versus urban patient population outcomes. The estimated median DFS for rural patients with early-stage breast cancer was 64 months (95% CI: 46–82) in comparison to 97 months (95% CI: 50–143) for urban patients ($p$ = 0.29) (Figure 4A). The estimated 10-year DFS was 33.7% in rural patients with early-stage invasive breast cancer compared to 50.3% in urban patients. Overall, 57 of 108 (53%) patients died during the follow-up period. The median OS of rural patients with all stages of breast cancer was 93 months (95% CI: 32–154) compared to 127 months (95% CI: 95–159) for their urban counterparts ($p$ = 0.63) (Figure 4B). The estimated 10-year OS of rural patients with all stages of breast cancer was 46.6% compared to 51% in urban patients. The median DFS of patients with early-stage invasive breast cancer diagnosed during the 2010–2019 period was 78 months (95% CI: 31–125) compared to 72 months (95% CI: 37–107) for patients diagnosed during the 2000–2009 period ($p$ = 0.47). The estimated median OS of patients with all stages of breast cancer diagnosed during the 2010–2019 period was 114 months

(95% CI: 95–132) compared to 108 months (95% CI: 48–168) for patients diagnosed during the 2000–2009 period (*p* = 0.70).

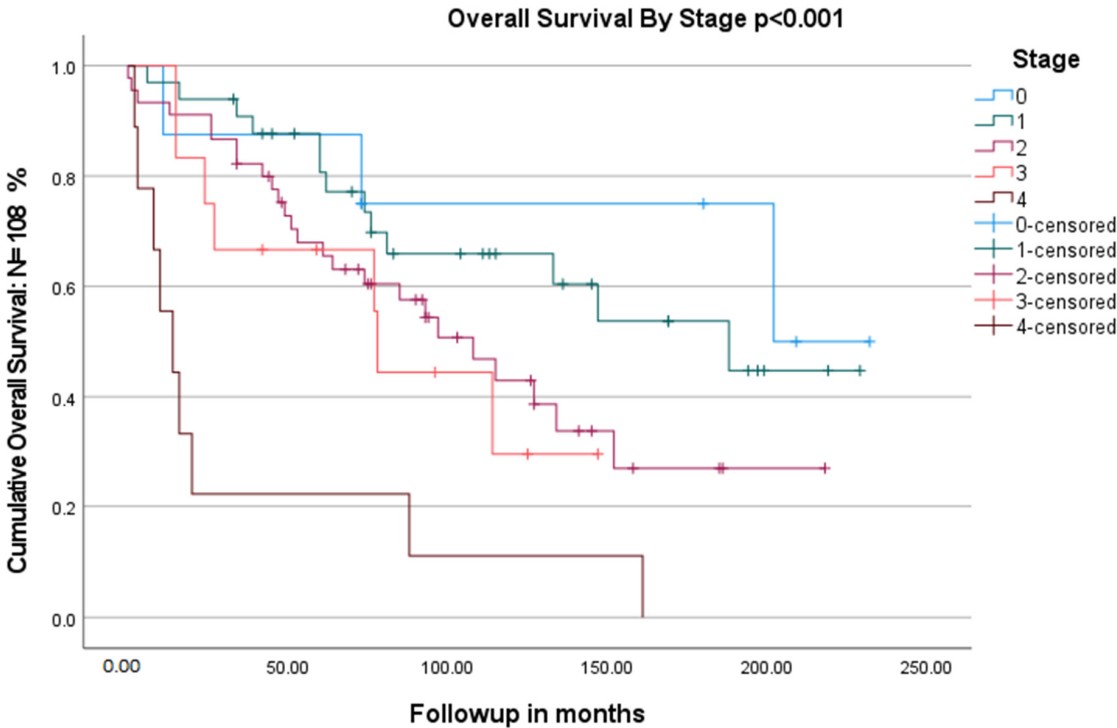

**Figure 2.** Kaplan–Meier curves for overall survival of 108 men based on stage of breast cancer at the time of diagnosis showed significant differences in overall survival (*p* < 0.001).

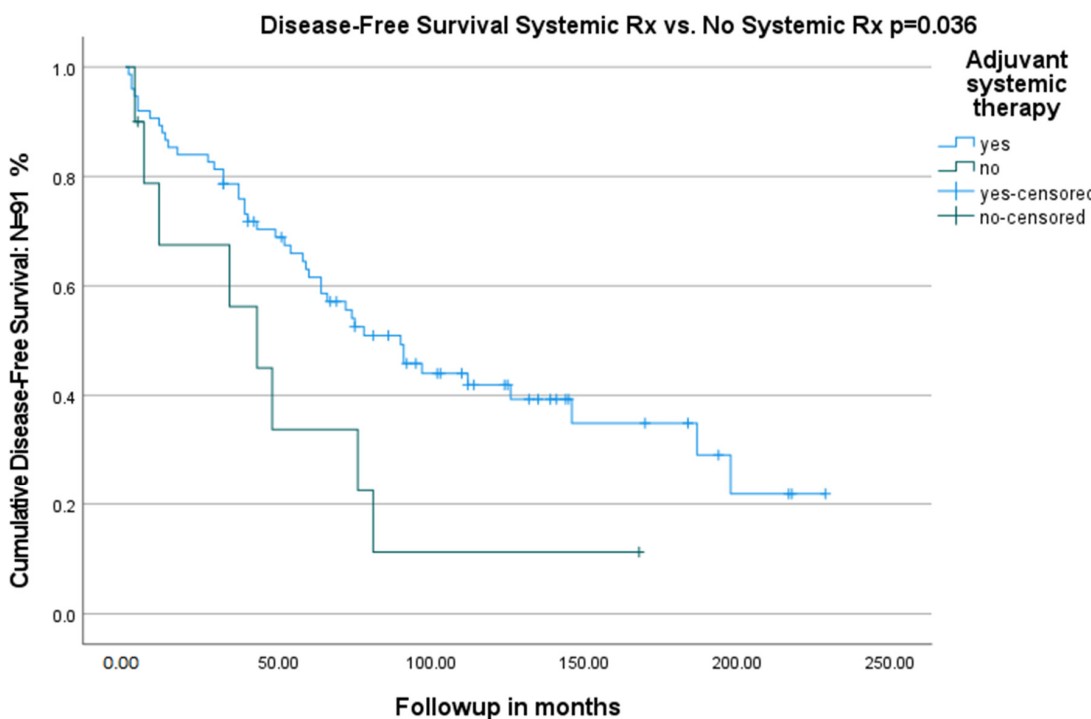

**Figure 3.** Kaplan–Meier disease-free survival curves based on adjuvant systemic therapy for 91 patients with early-stage invasive breast cancer that showed patients who received adjuvant systematic therapy have a significantly better disease-free survival (*p* = 0.036).

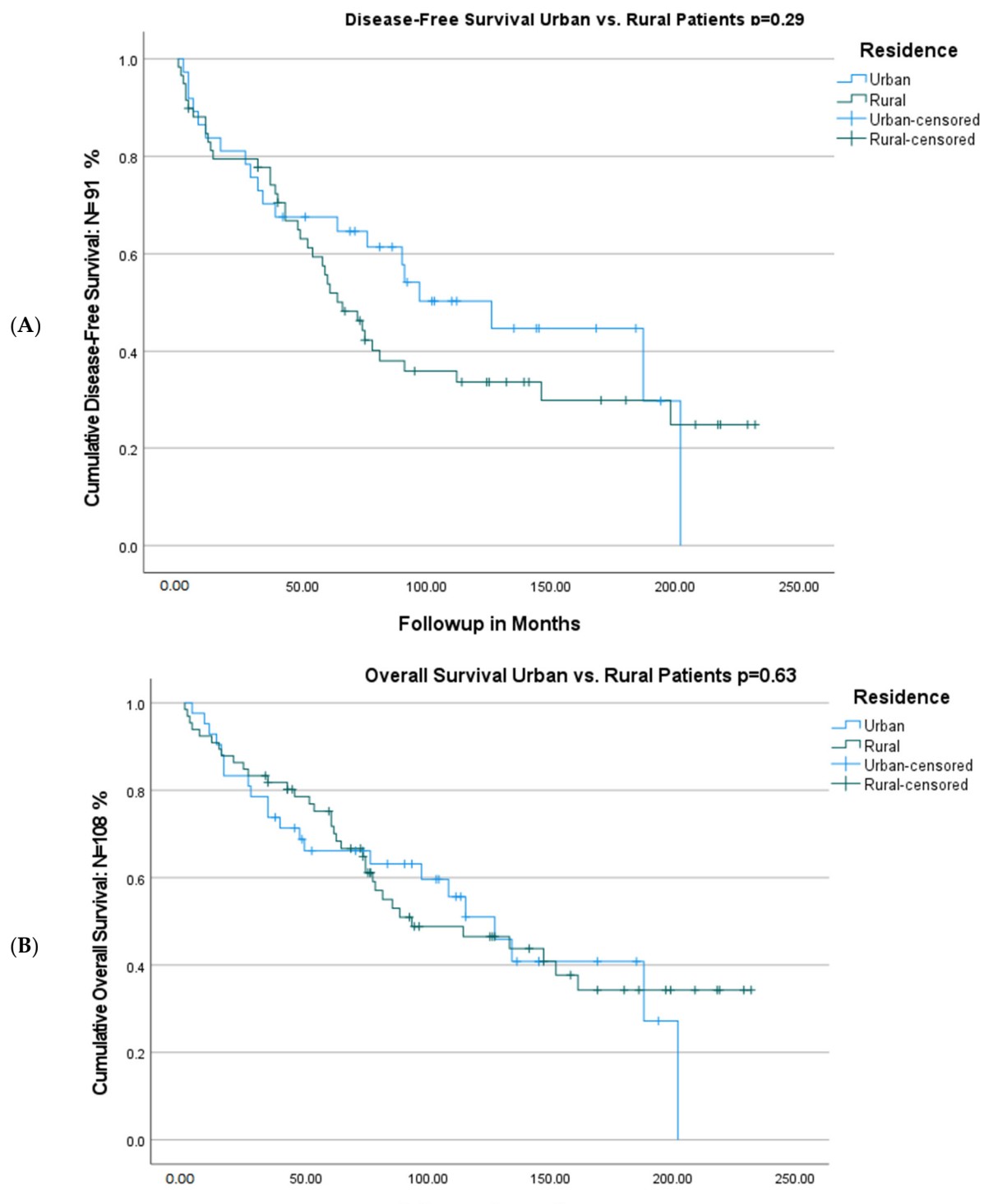

**Figure 4.** Comparison of rural versus urban patient outcomes. (**A**) Kaplan–Meier disease-free survival curves of 91 patients with early-stage invasive breast cancer showed no difference between urban and rural patient populations (*p* = 0.29). (**B**) Kaplan–Meier overall survival curves of 108 patients with breast cancer showed no difference between urban and rural patient populations (*p* = 0.63).

### 3.3. The Cox Proportional Analysis

The Cox proportional analysis was performed on the patient population to elucidate independent variables that have an effect on the outcome. The results are shown for variables that independently correlate with DFS and OS in Tables 3 and 4, respectively.

Rural residence was not independently correlated with DFS and OS. In addition, the time period of diagnosis (2000–2009) and/or elevated neutrophil:lymphocyte were not correlated with inferior outcomes (Tables 3 and 4). For patients with early-stage invasive breast cancer, multivariable analysis showed that WHO performance status $\geq$ 2 (HR 2.82 [95% CI: 1.14–6.94]), node-positive disease (HR 2.32 [95% CI: 1.22–4.40]), and not receiving adjuvant systemic therapy (HR 2.46 [1.03–5.92]) were significantly correlated with poorer DFS. For OS of patients with all stages of breast cancer, age $\geq$ 65 years (HR 2.39 [95% CI: 1.13–5.0]), stage IV disease (HR 7.82 [95% CI: 3.14–19.48]), and WHO performance status $\geq$ 2 (HR 4.35 [95% CI: 2.24–8.49]) were significantly correlated with inferior survival in multivariate analysis (Table 4).

**Table 3.** Cox proportional analysis for disease-free survival for 91 patients with early-stage invasive breast cancer.

| Variables | Univariate Analysis | | Multivariate Analysis | |
|---|---|---|---|---|
| | HR (CI) | *p* Value | HR (CI) | *p* Value |
| Age < 65 years | 1 | | 1 | |
| Age $\geq$ 65 years | 2.45 (1.27–4.72) | 0.007 | 1.32 (0.63–2.76) | 0.45 |
| Urban residence | 1 | | … … | |
| Rural residence | 1.35 (0.77–2.38) | 0.29 | … … | |
| Time period (2010–2019) | 1 | | … … | |
| Time period (2000–2009) | 1.22 (0.70–2.12) | 0.47 | … … | |
| No comorbid illness | 1 | | … … | |
| Comorbid illness | 1.15 (0.59–2.21) | 0.68 | … … | |
| No history of a secondary cancer | 1 | | … … | |
| Past history of a secondary cancer | 1.41 (0.80–2.47) | 0.23 | … … | |
| WHO performance status < 2 | 1 | | 1 | |
| WHO performance status $\geq$ 2 | 3.0 (1.44–8.32) | 0.003 | 2.82 (1.14–6.94) | 0.025 |
| $\leq$T2 disease | 1 | | … … | |
| >T2 disease | 1.42 (0.56–3.57) | 0.46 | … … | |
| Node negative disease | 1 | | 1 | |
| Node positive disease | 1.59 (0.91–2.76) | 0.10 | 2.32 (1.22–4.40) | 0.01 |
| Grade I or II tumor | 1 | | … … | |
| Grade III tumor | 1.26 (0.74–2.15) | 0.40 | … … | |
| Negative resection margin | 1 | | … … | |
| Positive resection margin | 1.58 (0.63–4.0) | 0.32 | … … | |
| HER2 negative disease | 1 | | 1 | |
| HER2 positive disease | 0.35 (1.09–1.14) | 0.08 | 0.38 (0.12–1.27) | 0.11 |
| Adjuvant systemic therapy | 1 | | 1 | |
| No adjuvant systemic therapy | 2.20 (1.03–4.71) | 0.04 | 2.47 (1.03–5.92) | 0.04 |
| Adjuvant chemotherapy | 1 | | … … | |
| No adjuvant chemotherapy | 1.29 (0.75–2.22) | 0.35 | … … | |

**Table 3.** *Cont.*

| Variables | Univariate Analysis | | Multivariate Analysis | |
|---|---|---|---|---|
| | HR (CI) | *p* Value | HR (CI) | *p* Value |
| Adjuvant radiation | 1 | | … … | |
| No adjuvant radiation | 0.72 (0.53–2.25) | 0.38 | … … | |
| Neutrophil:Lymphocyte ≤ 2.5 | 1 | | … … | |
| Neutrophil:Lymphocyte > 2.5 | 1.31 (0.75–2.28) | 0.34 | … … | |
| Alkaline phosphatase ≤ 110 U/L | 1 | | … … | |
| Alkaline phosphatase > 110 U/L | 1.72 (0.94–3.15) | 0.07 | … … | |
| Albumin ≥ 35 g/L | 1 | | … … | |
| Albumin < 35 g/L | 1.47 (0.68–3.15) | 0.32 | … … | |
| Creatinine < 105 µmol/L | 1 | | 1 | |
| Creatinine ≥ 105 µmol/L | 2.10 (1.21–3.64) | 0.008 | 1.94 (0.93–4.06) | 0.08 |
| Hemoglobin ≥ 135 g/L | 1 | | 1 | |
| Hemoglobin < 135 g/L | 1.72 (0.98–3.05) | 0.06 | 1.28 (0.52–3.14) | 0.59 |
| WBC > $11 \times 10^9$/L | 1 | | 1 | |
| WBC > $11 \times 10^9$/L | 1.69 (0.90–3.17) | 0.10 | 1.45 (0.56–3.75) | 0.44 |
| Platelets > $400 \times 10^9$/L | 1 | | … … | |
| Platelets > $400 \times 10^9$/L | 1.52 (0.79–2.91) | 0.21 | … … | |

WBC = white blood count; WHO = World Health Organization.

**Table 4.** Cox proportional analysis for overall survival for 108 patients with all stages of breast cancer.

| Variables | Univariate Analysis | | Multivariate Analysis | |
|---|---|---|---|---|
| | HR (95% CI) | *p* Value | HR (95% CI) | *p* Value |
| Age < 65 years | 1 | | 1 | |
| Age ≥ 65 years | 3.28 (1.68–6.39) | <0.001 | 2.37 (1.13–5.0) | 0.02 |
| Urban residence | 1 | | … … | |
| Rural residence | 0.95 (0.56–1.63) | 0.87 | ..... | |
| No comorbid illness | 1 | | … … | |
| Comorbid illness | 1.25 (0.65–2.40) | 0.50 | … … | |
| No history of a secondary cancer | 1 | | 1 | |
| Past history of a secondary cancer | 1.64 (0.92–2.90) | 0.09 | 1.69 (0.90-3.19) | 0.10 |
| Stage I to III disease | 1 | | 1 | |
| Stage IV disease | 4.78 (2.3–9.90) | <0.001 | 7.82 (3.14–19.48) | <0.001 |
| HER2 negative disease | 1 | | … … | |
| HER2 positive disease | 0.46 (0.14–1.50) | 0.20 | … … | |
| WHO performance status < 2 | 1 | | 1 | |
| WHO performance status ≥ 2 | 4.35 (2.24–8.49) | <0.001 | 3.25 (1.57–6.71) | 0.001 |
| Neutrophil:Lymphocyte ≤ 2.5 | 1 | | 1 | |
| Neutrophil:Lymphocyte > 2.5 | 1.66 (0.94–2.94) | 0.08 | 1.54 (0.82–2.91) | 0.18 |

**Table 4.** *Cont.*

| Variables | Univariate Analysis | | Multivariate Analysis | |
|---|---|---|---|---|
| | HR (95% CI) | *p* Value | HR (95% CI) | *p* Value |
| Time period (2010–2019) | 1 | | … … … | |
| Time period (2000–2009) | 1.11 (0.64–1.93) | 0.70 | … … … | |
| Alkaline phosphatase ≤ 110 | 1 | | 1 | |
| Alkaline phosphatase > 110 | 1.94 (1.15–3.30) | 0.014 | 2.13 (0.97–4.66) | 0.06 |
| Albumin ≥ 35 g/L | 1 | | … … … | |
| Albumin < 35 g/L | 1.34 (0.60–3.0) | 0.47 | … … … | |
| Creatinine < 105 | 1 | | 1 | |
| Creatinine ≥ 105 | 2.40 (1.42–4.10) | 0.001 | 1.74 (0.92–3.29) | 0.09 |
| Hemoglobin ≥ 135 g/L | 1 | | 1 | |
| Hemoglobin < 135 g/L | 2.0 (1.20–3.42) | 0.008 | 1.87 (0.88–3.95) | 0.10 |
| WBC ≤ $11 \times 10^9$/L | 1 | | … … … | |
| WBC > $11 \times 10^9$/L | 1.23 (0.68–2.22) | 0.50 | … … … | |
| Platelets ≤ $400 \times 10^9$/L | 1 | | … … … | |
| Platelets > $400 \times 10^9$/L | 1.47 (0.82–2.63) | 0.19 | … … … | |

WBC = white blood count; WHO = World Health Organization.

## 4. Discussion

Our results did not show differences in the outcomes of rural patients compared to their urban counterparts. Although numerical differences were observed for both DFS and OS, neither of them were statistically significant. Furthermore, after adjustment of other variables, the multivariable analysis did not show that rural residence was correlated with inferior DFS or OS. Several studies have reported inferior outcomes for rural women with breast cancer [11–15]; however, others have shown no differences between rural and urban breast cancer mortality outcomes [19]. One of the many reasons for reported disparity in breast cancer outcomes in rural settings has been attributed to diagnosis at later disease stage [12]. In our study cohort overall, 26% of rural men with breast cancer compared to 38% of urban men were diagnosed with stage I disease. Conversely, 15% of rural men compared to 5% of urban men with breast cancer at the time of diagnosis had stage III disease. However, the differences in the stage of the disease between urban and rural men with breast cancer were not statistically significant.

In our study cohort, almost all men with early-stage breast cancer underwent a mastectomy. We found that more than 90% of men had hormone receptor-positive breast cancer, which is consistent with prior findings [7,20–22]. For example, Johnson et al. reported a 93% prevalence of hormone receptor-positive breast cancer [20]. Another cohort study by Masci et al. found that about 97% of men with breast cancer had estrogen receptor-positive breast cancer [21]. Sarmiento et al. analyzed the National Cancer Database and reported that about 91% of male breast cancers were estrogen receptor-positive [22]. Triple-negative breast cancer observed in about 10% of female breast cancer was a relatively rare finding in our study cohort and only one patient had triple-negative breast cancer. Of note, unlike their female counterparts who have a high rate of stage I cancer at the time of diagnosis, most men were diagnosed with stage II breast cancer. The rate of diagnosis of stage III and de novo stage IV disease was similar between men and women. In addition, stage-based OS for this cohort was similar to those found in the literature for male breast cancer, with stage IV having the worst clinical outcomes and ductal carcinoma in situ (DCIS) having the best [22,23].

In multivariable analysis, older age, poor WHO performance status, and stage IV disease were correlated with poorer OS. Whereas for early-stage cancer, poor WHO performance status, node-positive disease, and lack of adjuvant systemic therapy were poorly correlated with DFS. Both older age and stage IV disease have been shown to decrease survival in male breast cancer populations [22,24–26]. Likewise, similar to females, lymph node-positive disease has been reported to be associated with inferior outcomes [23,25]. Poor performance status is known to be an important prognostic factor and, especially in advanced-stage disease, has been associated with inferior outcomes [27]. Our results showed that male patients with early-stage breast cancer and a WHO performance status of >1 had a significantly high risk of an event, independent of adjuvant systemic therapy. It is plausible that patients with borderline poor performance status may experience treatment delay or early discontinuation of treatment that may result in inferior outcomes [28,29]. Furthermore, lack of systemic therapy was strongly correlated with inferior outcomes in male breast cancer. It signifies the importance of adjuvant systemic therapy in male breast cancer; especially in our cohort, more than 50% of patients were diagnosed with stage II or III breast cancer. Adjuvant tamoxifen is recommended for hormone receptor-positive male breast cancer to reduce recurrent disease [30,31].

Despite the retrospective nature of the study, one of the key strengths of this multicenter population-based study is the lack of selection bias. All cases diagnosed in the entire province over a period of 20 years were included in this study. There was no loss to follow up. In addition, unlike many population-based studies using administrative data, individual medical records were reviewed and information about important clinical variables, including performance status and important laboratory values was collected. However, due to a relatively low sample size, the study does not have the power to detect small differences. Despite this limitation, due to the rarity of breast cancer in males, this study elucidates the prognostic factors that affect this unique group. Further research should be undertaken to determine whether the numerical differences seen in rural residence survival are observed in a larger patient population.

## 5. Conclusions

In summary, our results showed that although rural residence was not significantly correlated with outcomes, men with breast cancer and rural residence had numerically inferior survival. Poor performance status, node-positive disease, and lack of adjuvant systemic therapy were correlated with inferior outcomes in men with early-stage breast cancer.

**Author Contributions:** Conceptualization, L.A.B.F., H.I.C., D.L. and S.A.; Software, S.A.; Validation, S.A.; Formal analysis, S.A.; Investigation, L.A.B.F., O.A., H.I.C., R.D., A.E.-G., P.G., G.G., K.H., N.I., D.L., S.M., M.M., K.J., P.M., M.P., M.S., A.S., P.W., M.Y. and S.A.; Resources, S.A.; Data curation, L.A.B.F.; Writing—original draft, L.A.B.F. and S.A.; Writing—review and editing, O.A., H.I.C., R.D., A.E.-G., P.G., G.G., K.H., N.I., D.L., S.M., M.M., K.J., P.M., M.P., M.S., A.S., P.W., M.Y. and S.A.; Supervision, S.A.; Project administration, L.A.B.F.; Funding acquisition, L.A.B.F. and S.A. All authors have read and agreed to the published version of the manuscript.

**Funding:** This research was funded by the Mach-Gaensslen Foundation of Canada and the College of Medicine University of Saskatchewan, Dean Summer Project 2021-22.

**Institutional Review Board Statement:** The University of Saskatchewan Biomedical Research Ethics Board (Bio-REB) approved the study.

**Informed Consent Statement:** Patient consent was waived due to retrospective design of the study.

**Data Availability Statement:** Part of the data was presented at the 18th St. Gallen International Breast Cancer Conference in 15–18 March 2023, Vienna, Austria.

**Acknowledgments:** We are appreciative of the Mach-Gaensslen Foundation of Canada and the College of Medicine, University of Saskatchewan for supporting this research. We are very thankful to the Saskatchewan Cancer Agency for the support and access to the data.

**Conflicts of Interest:** The authors declare no conflict of interest.

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
