# Peer review of "Outcomes of Rural Men with Breast Cancer: A Multicenter Population Based Retrospective Cohort Study"

_cancers, doi:10.3390/cancers15071995_

Round 1

Reviewer 1 Report

Dear Sirs, I find the theme chosen very interesting, more so because I am researching the subject myself and I have published a literature review on male breast cancer. Congratulations on your study, I am sure that it has involved a significant effort, first of all,  in gathering all the data and , secondly, in doing the research itself! The major interest worldwide in breast cancer is mainly in its manifestation in  the female sex, due to its very high incidence and less in  rare forms, but I am sure that once more aspects related to male breast cancer are discovered, the interest in this matter will increase accordingly. 

Author Response

March 18, 2023

                                                                                                                                 Dr. Samuel C Mok                                                                                                  Editor-inChief                                                                                                            Cancers                                                                                      

RE: Cancers-2247337- Outcomes of Rural Men with Breast Cancer: A Multicenter Population Based Retrospective Cohort Study

Dear Dr. Mok,

Thank you very much for your correspondence about the above manuscript submitted for consideration for publication in Cancers. We appreciated the comments made by the reviewers and have thoroughly addressed them in the revision of the manuscript as described below:

Reviewers 1

Dear sir I find the theme chosen very interesting, more so because I am researching the subject myself and I have published a literature review on male breast cancer. Congratulations on your study, I am sure that it has involved a significant effort, first of all, in gathering all the data and, secondly, in doing the research itself! The major interest worldwide in breast cancer is mainly in its manifestation in the female sex, due to its very high incidence and less in rare forms, but I am sure that once more aspects related to male breast cancer are discovered, the interest in this matter will increase accordingly. 

Thanks very much. We appreciate the positive comments very much.

Reviewers 2

This is a retrospective cohort study involving men with breast cancer  (MBC) diagnosed in Canada in the period 2000-2019.  Compared with female BC, the male BC usually occurs later in life, with higher stage and lower grade and more often displays positive oestrogen and progesterone receptor status. Previous epidemiological collaborative studies defined clinical and pathological characteristics of MBC and the main predictors of specific outcomes.

The study of Fisher L et al. aimed to evaluate the survival outcomes of rural compared to urban patients in a series of 108 MBC (66 rural vs 42 urban) collected in the province of Saskatchewan (Canada) in the period 2000-2019. No significant differences on overall survival (OS) and disease free survival (DFS)  were found between rural and urban patients. Selected individual parameters emerged at multivariate Cox regression analysis as significant independent predictors of inferior survival outcomes.

Overall, the study is well defined. Although the sample size is quite small, the authors come to specific conclusions, defining some significant independent predictors of survival except the rural residence. However, some inconsistencies emerged, particularly in the statistical methods used and the presentation of results. Therefore, the manuscript needs to be revised.

Below there are some points that need to be clarified.

 Thanks very much for the positive comments and feedback.

Methods

- Line 85.  The definition of disease-free survival is not correct. The DFS is usually defined as the time from the date of surgery to the occurrence of specific events as recurrences of the disease, local relapses, distant metastases, or secondary cancers, but not death. The death is the final event for the definition of Overall Survival (OS), as the time from the diagnosis of the disease/date of surgery to the date of death from any cause or the last follow-up. Furthermore, in this series, the last follow-up is not defined.

Thanks very much for this comment. In most early stage-breast cancer clinical trials DFS refers to time from treatment until the recurrence of disease, a new primary invasive cancer or death after undergoing curative-intent treatment. For example, in HERA trial DFS was defined as time from randomization to the first occurrence of any of the following disease-free–survival events: recurrence of breast cancer at any site; the development of ipsilateral or contralateral breast cancer, including ductal carcinoma in situ but not lobular carcinoma in situ; second nonbreast malignant disease other than basal-cell or squamous-cell carcinoma of the skin or carcinoma in situ of the cervix; or death from any cause without documentation of a cancer-related event (Piccart-Gebhart et al). We have included relapse-free survival which is defined as the time from the date of surgery of early-stage operable breast cancer to the date of relapse or death from any cause.  We have added in information in the method and result section of the revised manuscript. In addition, we have provided follow up duration and date of last follow up (August 2022).

Piccart-Gebhart MJ, Procter M, Leyland-Jones B, Goldhirsch A, Untch M, Smith I, Gianni L, Baselga J, Bell R, Jackisch C, Cameron D, Dowsett M, Barrios CH, Steger G, Huang CS, Andersson M, Inbar M, Lichinitser M, Láng I, Nitz U, Iwata H, Thomssen C, Lohrisch C, Suter TM, Rüschoff J, Suto T, Greatorex V, Ward C, Straehle C, McFadden E, Dolci MS, Gelber RD; Herceptin Adjuvant (HERA) Trial Study Team. Trastuzumab after adjuvant chemotherapy in HER2-positive breast cancer. N Engl J Med. 2005 Oct 20;353(16):1659-72. 

Results

- line 109.  the identification of the study’s series reported in the text does not match with the flowchart reported in the figure 1;   

Thanks very much for identifying a typo error in the figure 1. There was only one patients with another active second malignancy at the time of breast cancer diagnosis that was excluded. We have made the correction in figure 1 of the revised paper.

- line 112/table 1.    1-Table 1 shows the baseline characteristics of the patients, but it must be clarified what is shown in each column of the table, for example number (N) and percentages between brackets (%), or mean and standard deviation (SD) ;    2- For a better understanding of the table, each categorical parameter reported must be listed in its     different levels (for example, smoking: current, former, never) with an exact percentage not considering the missing data;    3- to check the T status (1 missing data?) ;    4- what is the meaning of SCC?    5- to specify the unit of measure used for the laboratory variables (creatinine, albumin…);    6- to specify the statistical method used for p-value estimation; 

Thanks very much for the feedback.

1-Most of the data in the columns are number and % various as laboratory values are mean with standard deviation. We have clarified it in the revised table 1 where applicable.

2-Most of categorical variables are specified including stage, T status, grade, surgery, nodal status and hormone receptor status, etc. Smoking has been clarified as active smoker. Likewise presence of comorbid illnesses and a past history of secondary cancer have been clarified in the revised table 1.

3-T status information was missing from 1 patient that has been specified in the footnote of the revised table 1.

4-SCC was referred to a center. We have omitted this information from the revised table 1.

5- All units have been added in the revised table 1.

6.- Statistical methods have been clarified in the section of statistical analysis that chi-square and Fisher Exact tests were performed for p values estimation.

- line 134/table 2.    1- to report in the text the exact number of events for survival analysis (deaths and recurrences);    2- to specify what is shown in each column of the table (to see above, for table 1) ;    3- to better define the parameters reported in the table;    4- to specify the statistical method used for p-value estimation;  

1-number of deaths are included in the text of the revised paper.

2-% and range are specified in the revised table 2 as applicable.

3- a definition section has been added in revised paper to clarify various terms used in the study.

4-In statistical analysis section methods for estimation of p values for continuous and categorical variable have been defined.   

- line 145/figure 2.    1-  to report in the text and also in the figure 2 the p-value from log rank test for the OS           based on stage;    2- to add the unit of measure (%)  used for cumulative survival (OS) in the y-axis of the graph;    3- to specify in the legend of the figure the number of patients evaluated  in the survival analysis;   

1-P value was added in the result section and revised figure 2.

2-% and number of patients have been added in the y-axis of the figure 2.

3-number of patients have been added.

- line 151/figure 3.    1- to report in the figure 3 the p-value from log rank test for the DFS (p=0.036);    2- to correct the definition of the y-axis of the graph (DFS (%));    3- to specify in the legend of the figure the number of patients evaluated  in the survival analysis;   

1-P value has been added in the figure and figure legend.

2--% and number of patients have been added in the y-axis of the figure 3.

3-number of patients have been added in figure and figure legend.  

- line 152/figure 4.    1- to report in the figure 4A the p-value from log rank test for the DFS (p=0.29);    2- to correct the definition of the y-axis of the graph (DFS (%));    3- to report in the figure 4B the p-value from log rank test for the OS (p=0.63);    4- to correct the definition of the y-axis of the graph (OS (%));   

1-P value has been added in the figure 4.

2-% and number of patients have been added in the y-axis of the figure 3.

3- P value has been added in the  figure 4.

4--% and number of patients have been added in the y-axis of the figure 4.

line 176    table 3,   Cox regression analysis for DFS; 1-      to define the reference category for each parameter evaluated; 2-      to specify the uni-variate regression model and the multi-variate regression model; 3-      to uniform the term “grade” with table 1 (3, III ?); 4-      the parameter “secondary cancer“ must not be entered in the model because it is among the “events” evaluated in the DFS; 5-      to specify in the legend of the table the exact number of patients evaluated for the DFS analysis;   

1-reference categories are added in the revised table 3 and 4.

2-Uni and maultivariate analyses have been clarified.

3-uniform term has been used for grading.

4-Seconday cancer was added as event if patient developed a new primary cancer during the follow period. Past history of secondary cancer was not considered an event was has been specified in the table and text.

5- numbers have been specified in the table legend.

table 4,   Cox regression analysis for OS; 1-      to define the reference category for each parameter evaluated; 2-      to specify in the legend of the table the exact number of patients evaluated for the OS analysis;     - line 263, 

1-reference categories have been added.

2- numbers are specified in the table  legend

references 1- to check the references n. 6, 17, 23 – to define the exact number of volume and/or pages.

1-references have been corrected.

In addition as per editorial comments we have expanded the text about 3,000 words and added 5 new references with total references of 31.

In summary, we appreciate the helpful comments provided by the reviewers and hope that the revised manuscript is now suitable for publication in the Cancers.

Sincerely,

Shahid Ahmed, MD, PhD, FRCPC, FRCP (Edin), FACP

Professor of Oncology

Saskatchewan Cancer Agency

University of Saskatchewan

20 Campus Drive

Saskatoon, SK

Canada, S7N4H4

Reviewer 2 Report

Comments on paper of Fisher L. et al. “Outcomes of rural men with breast cancer: a multicenter population based retrospective cohort study”

This is a retrospective cohort study involving men with breast cancer  (MBC) diagnosed in Canada in the period 2000-2019.  Compared with female BC, the male BC usually occurs later in life, with higher stage and lower grade and more often displays positive oestrogen and progesterone receptor status. Previous epidemiological collaborative studies defined clinical and pathological characteristics of MBC and the main predictors of specific outcomes.

The study of Fisher L et al. aimed to evaluate the survival outcomes of rural compared to urban patients in a series of 108 MBC (66 rural vs 42 urban) collected in the province of Saskatchewan (Canada) in the period 2000-2019. No significant differences on overall survival (OS) and disease free survival (DFS)  were found between rural and urban patients. Selected individual parameters emerged at multivariate Cox regression analysis as significant independent predictors of inferior survival outcomes.

Overall, the study is well defined. Although the sample size is quite small, the authors come to specific conclusions, defining some significant independent predictors of survival except the rural residence. However, some inconsistencies emerged, particularly in the statistical methods used and the presentation of results. Therefore, the manuscript needs to be revised.

Below there are some points that need to be clarified.

Methods

- Line 85.  The definition of disease-free survival is not correct. The DFS is usually defined as the time from the date of surgery to the occurrence of specific events as recurrences of the disease, local relapses, distant metastases, or secondary cancers, but not death. The death is the final event for the definition of Overall Survival (OS), as the time from the diagnosis of the disease/date of surgery to the date of death from any cause or the last follow-up. Furthermore, in this series, the last follow-up is not defined.

Results

- line 109.  the identification of the study’s series reported in the text does not match with the flowchart reported in the figure 1;   - line 112/table 1.    1-Table 1 shows the baseline characteristics of the patients, but it must be clarified what is shown in each column of the table, for example number (N) and percentages between brackets (%), or mean and standard deviation (SD) ;    2- For a better understanding of the table, each categorical parameter reported must be listed in its     different levels (for example, smoking: current, former, never) with an exact percentage not considering the missing data;    3- to check the T status (1 missing data?) ;    4- what is the meaning of SCC?    5- to specify the unit of measure used for the laboratory variables (creatinine, albumin…);    6- to specify the statistical method used for p-value estimation;   - line 134/table 2.    1- to report in the text the exact number of events for survival analysis (deaths and recurrences);    2- to specify what is shown in each column of the table (to see above, for table 1) ;    3- to better define the parameters reported in the table;    4- to specify the statistical method used for p-value estimation;   - line 145/figure 2.    1-  to report in the text and also in the figure 2 the p-value from log rank test for the OS           based on stage;    2- to add the unit of measure (%)  used for cumulative survival (OS) in the y-axis of the graph;    3- to specify in the legend of the figure the number of patients evaluated  in the survival analysis;   - line 151/figure 3.    1- to report in the figure 3 the p-value from log rank test for the DFS (p=0.036);    2- to correct the definition of the y-axis of the graph (DFS (%));    3- to specify in the legend of the figure the number of patients evaluated  in the survival analysis;   - line 152/figure 4.    1- to report in the figure 4A the p-value from log rank test for the DFS (p=0.29);    2- to correct the definition of the y-axis of the graph (DFS (%));    3- to report in the figure 4B the p-value from log rank test for the OS (p=0.63);    4- to correct the definition of the y-axis of the graph (OS (%));   - line 176    table 3,   Cox regression analysis for DFS; 1-      to define the reference category for each parameter evaluated; 2-      to specify the uni-variate regression model and the multi-variate regression model; 3-      to uniform the term “grade” with table 1 (3, III ?); 4-      the parameter “secondary cancer“ must not be entered in the model because it is among the “events” evaluated in the DFS; 5-      to specify in the legend of the table the exact number of patients evaluated for the DFS analysis;      table 4,   Cox regression analysis for OS; 1-      to define the reference category for each parameter evaluated; 2-      to specify in the legend of the table the exact number of patients evaluated for the OS analysis;     - line 263, references 1- to check the references n. 6, 17, 23 – to define the exact number of volume and/or pages.

Author Response

March 18, 2023

                                                                                                                                Dr. Samuel C. Mok                                                                                                  Editor-in Chief                                                                                                          Cancers                                                                                      

RE: Cancers-2247337- Outcomes of Rural Men with Breast Cancer: A Multicenter Population Based Retrospective Cohort Study

Dear Dr. Mok,

Top of Form

Thank you very much for your correspondence about the above manuscript submitted for consideration for publication in Cancers. We appreciated the comments made by the reviewers and have thoroughly addressed them in the revision of the manuscript as described below:

Reviewers 1

Dear sir I find the theme chosen very interesting, more so because I am researching the subject myself and I have published a literature review on male breast cancer. Congratulations on your study, I am sure that it has involved a significant effort, first of all, in gathering all the data and, secondly, in doing the research itself! The major interest worldwide in breast cancer is mainly in its manifestation in the female sex, due to its very high incidence and less in rare forms, but I am sure that once more aspects related to male breast cancer are discovered, the interest in this matter will increase accordingly. 

Thanks very much. We appreciate the positive comments very much.

Reviewers 2

This is a retrospective cohort study involving men with breast cancer  (MBC) diagnosed in Canada in the period 2000-2019.  Compared with female BC, the male BC usually occurs later in life, with higher stage and lower grade and more often displays positive oestrogen and progesterone receptor status. Previous epidemiological collaborative studies defined clinical and pathological characteristics of MBC and the main predictors of specific outcomes.

The study of Fisher L et al. aimed to evaluate the survival outcomes of rural compared to urban patients in a series of 108 MBC (66 rural vs 42 urban) collected in the province of Saskatchewan (Canada) in the period 2000-2019. No significant differences on overall survival (OS) and disease free survival (DFS)  were found between rural and urban patients. Selected individual parameters emerged at multivariate Cox regression analysis as significant independent predictors of inferior survival outcomes.

Overall, the study is well defined. Although the sample size is quite small, the authors come to specific conclusions, defining some significant independent predictors of survival except the rural residence. However, some inconsistencies emerged, particularly in the statistical methods used and the presentation of results. Therefore, the manuscript needs to be revised.

Below there are some points that need to be clarified.

 Thanks very much for the positive comments and feedback.

Methods

- Line 85.  The definition of disease-free survival is not correct. The DFS is usually defined as the time from the date of surgery to the occurrence of specific events as recurrences of the disease, local relapses, distant metastases, or secondary cancers, but not death. The death is the final event for the definition of Overall Survival (OS), as the time from the diagnosis of the disease/date of surgery to the date of death from any cause or the last follow-up. Furthermore, in this series, the last follow-up is not defined.

Thanks very much for this comment. In most early stage-breast cancer clinical trials DFS refers to time from treatment until the recurrence of disease, a new primary invasive cancer or death after undergoing curative-intent treatment. For example, in HERA trial DFS was defined as time from randomization to the first occurrence of any of the following disease-free–survival events: recurrence of breast cancer at any site; the development of ipsilateral or contralateral breast cancer, including ductal carcinoma in situ but not lobular carcinoma in situ; second nonbreast malignant disease other than basal-cell or squamous-cell carcinoma of the skin or carcinoma in situ of the cervix; or death from any cause without documentation of a cancer-related event (Piccart-Gebhart et al). We have included relapse-free survival which is defined as the time from the date of surgery of early-stage operable breast cancer to the date of relapse or death from any cause.  We have added in information in the method and result section of the revised manuscript. In addition, we have provided follow up duration and date of last follow up (August 2022).

Piccart-Gebhart MJ, Procter M, Leyland-Jones B, Goldhirsch A, Untch M, Smith I, Gianni L, Baselga J, Bell R, Jackisch C, Cameron D, Dowsett M, Barrios CH, Steger G, Huang CS, Andersson M, Inbar M, Lichinitser M, Láng I, Nitz U, Iwata H, Thomssen C, Lohrisch C, Suter TM, Rüschoff J, Suto T, Greatorex V, Ward C, Straehle C, McFadden E, Dolci MS, Gelber RD; Herceptin Adjuvant (HERA) Trial Study Team. Trastuzumab after adjuvant chemotherapy in HER2-positive breast cancer. N Engl J Med. 2005 Oct 20;353(16):1659-72. 

Results

- line 109.  the identification of the study’s series reported in the text does not match with the flowchart reported in the figure 1;   

Thanks very much for identifying a typo error in the figure 1. There was only one patients with another active second malignancy at the time of breast cancer diagnosis that was excluded. We have made the correction in figure 1 of the revised paper.

- line 112/table 1.    1-Table 1 shows the baseline characteristics of the patients, but it must be clarified what is shown in each column of the table, for example number (N) and percentages between brackets (%), or mean and standard deviation (SD) ;    2- For a better understanding of the table, each categorical parameter reported must be listed in its     different levels (for example, smoking: current, former, never) with an exact percentage not considering the missing data;    3- to check the T status (1 missing data?) ;    4- what is the meaning of SCC?    5- to specify the unit of measure used for the laboratory variables (creatinine, albumin…);    6- to specify the statistical method used for p-value estimation; 

Thanks very much for the feedback.

1-Most of the data in the columns are number and % various as laboratory values are mean with standard deviation. We have clarified it in the revised table 1 where applicable.

2-Most of categorical variables are specified including stage, T status, grade, surgery, nodal status and hormone receptor status, etc. Smoking has been clarified as active smoker. Likewise presence of comorbid illnesses and a past history of secondary cancer have been clarified in the revised table 1.

3-T status information was missing from 1 patient that has been specified in the footnote of the revised table 1.

4-SCC was referred to a center. We have omitted this information from the revised table 1.

5- All units have been added in the revised table 1.

6.- Statistical methods have been clarified in the section of statistical analysis that chi-square and Fisher Exact tests were performed for p values estimation.

- line 134/table 2.    1- to report in the text the exact number of events for survival analysis (deaths and recurrences);    2- to specify what is shown in each column of the table (to see above, for table 1) ;    3- to better define the parameters reported in the table;    4- to specify the statistical method used for p-value estimation;  

1-number of deaths are included in the text of the revised paper.

2-% and range are specified in the revised table 2 as applicable.

3- a definition section has been added in revised paper to clarify various terms used in the study.

4-In statistical analysis section methods for estimation of p values for continuous and categorical variable have been defined.   

- line 145/figure 2.    1-  to report in the text and also in the figure 2 the p-value from log rank test for the OS           based on stage;    2- to add the unit of measure (%)  used for cumulative survival (OS) in the y-axis of the graph;    3- to specify in the legend of the figure the number of patients evaluated  in the survival analysis;   

1-P value was added in the result section and revised figure 2.

2-% and number of patients have been added in the y-axis of the figure 2.

3-number of patients have been added.

- line 151/figure 3.    1- to report in the figure 3 the p-value from log rank test for the DFS (p=0.036);    2- to correct the definition of the y-axis of the graph (DFS (%));    3- to specify in the legend of the figure the number of patients evaluated  in the survival analysis;   

1-P value has been added in the figure and figure legend.

2--% and number of patients have been added in the y-axis of the figure 3.

3-number of patients have been added in figure and figure legend.  

- line 152/figure 4.    1- to report in the figure 4A the p-value from log rank test for the DFS (p=0.29);    2- to correct the definition of the y-axis of the graph (DFS (%));    3- to report in the figure 4B the p-value from log rank test for the OS (p=0.63);    4- to correct the definition of the y-axis of the graph (OS (%));   

1-P value has been added in the figure 4.

2-% and number of patients have been added in the y-axis of the figure 3.

3- P value has been added in the  figure 4.

4--% and number of patients have been added in the y-axis of the figure 4.

line 176    table 3,   Cox regression analysis for DFS; 1-      to define the reference category for each parameter evaluated; 2-      to specify the uni-variate regression model and the multi-variate regression model; 3-      to uniform the term “grade” with table 1 (3, III ?); 4-      the parameter “secondary cancer“ must not be entered in the model because it is among the “events” evaluated in the DFS; 5-      to specify in the legend of the table the exact number of patients evaluated for the DFS analysis;   

1-reference categories are added in the revised table 3 and 4.

2-Uni and maultivariate analyses have been clarified.

3-uniform term has been used for grading.

4-Seconday cancer was added as event if patient developed a new primary cancer during the follow period. Past history of secondary cancer was not considered an event was has been specified in the table and text.

5- numbers have been specified in the table legend.

table 4,   Cox regression analysis for OS; 1-      to define the reference category for each parameter evaluated; 2-      to specify in the legend of the table the exact number of patients evaluated for the OS analysis;     - line 263, 

1-reference categories have been added.

2- numbers are specified in the table  legend

references 1- to check the references n. 6, 17, 23 – to define the exact number of volume and/or pages.

1-references have been corrected.

In addition as per editorial comments we have expanded the text about 3,000 words and added 5 new references with total references of 31.

In summary, we appreciate the helpful comments provided by the reviewers and hope that the revised manuscript is now suitable for publication in the Cancers.

Sincerely,

Shahid Ahmed, MD, PhD, FRCPC, FRCP (Edin), FACP

Professor of Oncology

Saskatchewan Cancer Agency

University of Saskatchewan

20 Campus Drive

Saskatoon, SK

Canada, S7N4H4
